# Association of *Euterpe oleracea, Bixa orellana, Myciaria dubia,* and *Astrocaryum aculeatum* (the Terasen® Nutraceutical) Increases the Lifespan of *Caenorhabditis elegans*

Ester Lopes de Melo [1,2], Bruno Augusto Machado Tavares [1], Nayara Nílcia Dias Colares [1],
Aline Lopes do Nascimento [1,3], Helison de Oliveira Carvalho [1], Andrés Navarrete Castro [4],
Arlindo César Matias Pereira [5], Carlos Eduardo Winter [6], Iracirema da Silva Sena [1],
Luiz Fernando Moreira [1] and José Carlos Tavares Carvalho [1,2,3,*]

1   Drug Research Laboratory, Department of Biological and Health Sciences, Federal University of Amapá,
    CEP, Macapá 68903-419, AP, Brazil; esterlpsmelo@hotmail.com (E.L.d.M.); brunot44435@gmail.com (B.A.M.T.);
    nayarandiascolares@gmail.com (N.N.D.C.); alinelopes1717@gmail.com (A.L.d.N.);
    helison_farma@hotmail.com (H.d.O.C.); ciremasena@gmail.com (I.d.S.S.); contato@consulfarma.com (L.F.M.)
2   Postgraduate Program in Tropical Biodiversity, Department of Biological and Health Sciences,
    Federal University of Amapá, CEP, Macapá 68903-419, AP, Brazil
3   Postgraduate Program in Pharmaceutical Innovation, Department of Biological Sciences and Health,
    Federal University of Amapá, CEP, Macapá 68903-419, AP, Brazil
4   Laboratory of Natural Product Pharmacology, Department of Pharmacy, Faculty of Chemistry,
    National Autonomous University of Mexico, University City, Coyoacán, Mexico City 04510, Mexico;
    anavarrt@unam.mx
5   Faculty of Pharmaceutical Sciences of Ribeirao Preto, University of Sao Paulo,
    Ribeirão Preto 14040-020, SP, Brazil; arlindo.bio@outlook.com
6   Laboratory of Molecular Biology of Nematodes, Department of Parasitology, Institute of Biomedical Sciences,
    University of São Paulo, São Paulo 14040-020, SP, Brazil; cewinter@icb.usp.br
*   Correspondence: farmacos@unifap.br

**Abstract:** Aging is a complex process associated with tissue degeneration and an increased risk of age-related diseases. This study aimed to evaluate the impact of Terasen®, a nutraceutical containing standardized extracts of *Euterpe oleracea, Myrciaria dubia,* and purified oil of *Bixa orellana* and *Astrocaryum aculeatum* on the lifespan of *Caenorhabditis elegans*. The findings demonstrated that Terasen® exhibited significant radical scavenging in vitro, decreased the feeding behavior of *C. elegans* without affecting the animals' final size, increased the eggs laid in a concentration-dependent fashion, although the total progeny was reduced compared to the control, and increased the median and maximum lifespan. These findings suggest that Terasen® may improve the lifespan in *C. elegans*, warranting further investigation.

**Keywords:** longevity; healthspan; lifespan; nutraceuticals; *C. elegans*

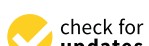



## 1. Introduction

Aging is a natural process during an individual's lifetime that leads to a decline in natural body functions [1,2]. The functional changes associated with aging have been identified as risk factors for various health problems, including hypertension, osteoporosis, diabetes, cataracts, heart failure, and neurodegenerative diseases [3].

According to the World Health Organization (WHO), global life expectancy is increasing, and it is projected that the elderly population will reach 1.4 billion people by 2030 and exceed 2.1 billion by 2050 [4]. With this demographic shift, it is imperative to seek solutions that address the detrimental effects of aging, ensuring that individuals can maintain their independence and experience healthy aging [2,5].

Pharmacological interventions aimed at prolonging human longevity and preventing age-related diseases are under investigation. The nematode *Caenorhabditis elegans* has

emerged as a promising model for studying the determinants of longevity. This tiny nematode measures around 1 mm in length as an adult. It is cultured in petri dishes using solid and liquid culture media, and its primary food source is *Escherichia coli*. With its two types of sex, hermaphrodite (XX) and male (XO), the hermaphrodite can generate 300 to 1400 offspring, facilitating the creation of a uniform genetic lineage [6].

Despite its simple anatomy with few tissues and organs, the nervous system, gastrointestinal tract, gonads, and muscles of *C. elegans* are comparable to those of more complex animals [7]. Although it lacks adipose tissue or a liver, its intestine can perform similar functions, such as lipid synthesis and lipoprotein secretion [8]. The genetic sequence of *C. elegans* is well known, and genomic research has revealed that its genome has 60 to 80% of genes similar to those of humans involved in diseases [9].

*C. elegans* has a short lifespan, compact size, and conserved genetic pathways that regulate aging, such as insulin signaling, oxidative stress response, and longevity-related genes [10].

Terasen® is a nutraceutical formulation composed of standardized extracts of *Euterpe oleracea*, *Myrciaria dubia*, and purified oil of *Bixa orellana* and *Astrocaryum aculeatum*. The phytochemical markers of Terasen® include anthocyanins, quercetin, polyphenols, carotenoids, geranylgeraniol, and tocotrienols, which have been extensively studied for their pharmacological properties and their potential impact on longevity. In this study, we employed the *C. elegans* model to investigate the effects of Terasen® on longevity.

## 2. Materials and Methods

### 2.1. Product

The nutraceutical product in the form of encapsulated granules, Terasen®, was provided by Ages Bioactive Compounds (São Paulo, SP, Brazil), batch URU201101.

### 2.2. Evaluation of Radical Scavenging Activity In Vitro

The Terasen® sample was extracted using DMSO and vortex agitation for the radical scavenging analysis. At the end of the procedure, all samples were prepared at a final concentration of 1 mg/mL. Gallic acid and DMSO were used as the standard and negative control, respectively. The results were expressed as the percentage of radical scavenging using the following equation: Inhibition (%) = $[(Ac - As)/Ac] \times 100$, where Ac is the absorbance of the negative control (DMSO) and as is the absorbance of the sample. All analyses were performed in triplicates.

#### 2.2.1. DPPH Radical Scavenging Activity

The method described in [11], with some modifications, was used to determine the radical scavenging activity through the DPPH method. The DPPH solution (2,2-diphenyl-1-picrylhydrazyl) was prepared by dissolving 1 mg of DPPH in 12 mL of absolute ethanol. Then, 270 µL of this solution and 30 µL of ethanol were added to the microplate, adjusting the solution with ethanol to achieve an absorbance of $1.00 \pm 0.1$ nm. For the test, 30 µL of the sample and 270 µL of the DPPH solution were added to the plate, and it was incubated in the dark for 30 min. After this time, the absorbance was measured at 517 nm using an ELISA reader (Kasuaki, Dr-200Bn-Bi, Baden-Württemberg, Germany).

#### 2.2.2. ABTS Radical Scavenging Assay

The radical scavenging activity was evaluated also using the ABTS method described by [12] with some modifications. The ABTS solution (7 mM) (2,2′-azinobis(3-ethylbenzothiazoline-6-sulfonic acid)) was mixed with a potassium persulfate solution (2.45 mM) and incubated at room temperature in the dark for 16 h. Then, 270 µL of the ABTS solution was mixed with 30 µL of the sample. The plate was incubated in the dark for 30 min, and the absorbance was measured at 630 nm using an ELISA reader (Kasuaki, Dr-200Bn-Bi).

*2.3. Caenorhabditis elegans Strains*

The N2 Bristol (wild-type) strains of *Caenorhabditis elegans* used in this study were obtained from the Department of Parasitology/ICB/USP (ICB, University of São Paulo, SP, Brazil). All nematodes were incubated at 20 °C and grown on NGM (nematode growth medium) seeded with *Escherichia coli* OP50 as a food source.

### 2.3.1. Reproduction Assay

The reproduction assay was conducted following the method described by [13]. Synchronized L4 nematodes ($n = 5$) were daily transferred until the end of the reproductive period to NGM plates with only *E. coli* OP50 in LB medium (negative control) or *E. coli* in LB medium plus different concentrations of Terasen® solubilized in the medium (250, 500, or 1000 µg/mL), and the eggs were counted.

### 2.3.2. Pharyngeal Pumping Rate

The pharyngeal pumping rate was evaluated using the methodology proposed by [14]. Synchronized L4 nematodes were raised on NGM plates and treated with only *E. coli* OP50 in LB medium (negative control) or *E. coli* in LB medium plus different concentrations of Terasen® solubilized in the medium (250, 500, or 1000 µg/mL). On the 3rd, 6th, and 9th day of adulthood, 10 worms were randomly selected, and the number of pharyngeal contractions during a 60 s interval was quantified. The experiment was performed three times per group, and the averaged values were analyzed.

### 2.3.3. Growth Alteration Assay

The growth alteration assay was performed following the method described by [15]. The animals were raised from the L1 stage on NGM plates receiving only *E. coli* OP50 in LB medium (negative control) or *E. coli* in LB medium plus different concentrations of Terasen® solubilized in the medium (250, 500, or 1000 µg/mL). On the 4th and 8th day of adulthood, the animals ($n = 10$) were photographed using a stereomicroscope and camera (Luxeo 4D, Labomed, CA, USA), and their body length was measured from head to tail using ImageJ software (v1.53u, Boston, MA, USA). The experiment was performed in triplicate.

### 2.3.4. Locomotion Analysis Assay

The locomotion analysis assay followed the protocol described by [16]. Initially, synchronized N2 nematodes were transferred to NGM plates and treated with only *E. coli* OP50 in LB medium (negative control) or *E. coli* in LB medium plus different concentrations of Terasen® solubilized in the medium (250, 500, or 1000 µg/mL). After 4 and 8 days of treatment, each nematode ($n = 6$) was placed on a glass dish with 100 µL of M9 buffer. After 1 min of recovery, the total number of body bends was counted over 20 s using a stereomicroscope.

### 2.3.5. Lifespan Assessment

The lifespan of nematodes was assessed following the method previously described [14]. Synchronized L4 stage nematodes ($n = 60$) were transferred to NGM plates and treated with only *E. coli* OP50 in LB medium (negative control) or *E. coli* in LB medium plus different concentrations of Terasen® solubilized in the medium (250, 500, or 1000 µg/mL). The nematodes were transferred daily to fresh NGM plates with their respective treatments for the first six days, followed by transfers every two days afterward. Daily worm counts were conducted until the death of all nematodes, with worms unresponsive to a gentle touch and a platinum wire marked as dead and excluded from the plates. The results were reported as the percentage of survival, mean lifespan, and median lifespan. The lifespan tests were conducted at a constant temperature of 20 °C.

*2.4. Statistical Analysis*

Statistical analysis was conducted using GraphPad Prism software (version 5.03). The results were presented as mean ± SEM. Two-Way ANOVA was used to compare data (factors: treatment; days), followed by multiple Tukey's post hoc tests in case of statistical difference. One-way ANOVA followed by Tukey's test was employed to compare area under curves (AUCs). A log-rank test was used to compare survival curves. Statistical significance was defined as $p < 0.05$.

## 3. Results

*3.1. Radical Scavenging Assessment*

In the DPPH and ABTS radical inhibition assays, the values obtained with Terasen® were 20.85 ± 2.99% and 69.95 ± 5.39% inhibition relative to DMSO, respectively, compared to 75.35 ± 4.42% and 95.82 ± 0.39% inhibition relative to DMSO in the gallic acid control (Figure 1). It was observed that the inhibition percentage was significantly higher in the ABTS assay, and this result is directly related to the phenolic compound content present in Terasen®. Phenolic compounds have a higher affinity for scavenging the ABTS radical due to differences in their chemical structures and the properties of free radicals [17,18].

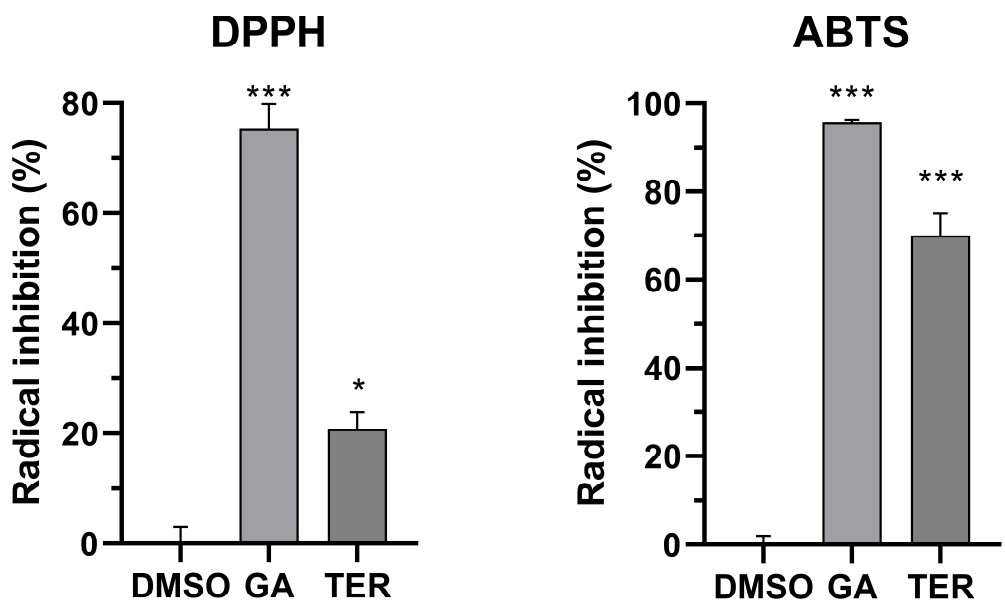

**Figure 1.** Radical scavenging activity of Terasen® (TER) in the DPPH and ABTS assay. Gallic Acid (GA) is shown as a reference compound. * and **** represent statistical differences compared to DMSO ($p < 0.05$ and 0.001; ANOVA followed by Tukey's posthoc test).

*3.2. Assessment of Pharyngeal Pumping Rate*

To investigate whether exposure to Terasen® could affect the pharyngeal pumping rate, a crucial measure for evaluating feeding behavior in *C. elegans* [15], an investigation was conducted in *C. elegans* exposed to different concentrations of Terasen® (250, 500, or 1000 µg/mL). Compared to the control group, there was a statistically significant reduction in pharyngeal pumping rate on days 3, 6, and 9 of adult life in *C. elegans* exposed to Terasen® ($p < 0.05$; Figure 2). These findings suggest that exposure to Terasen® may reduce the feeding behavior of these organisms.

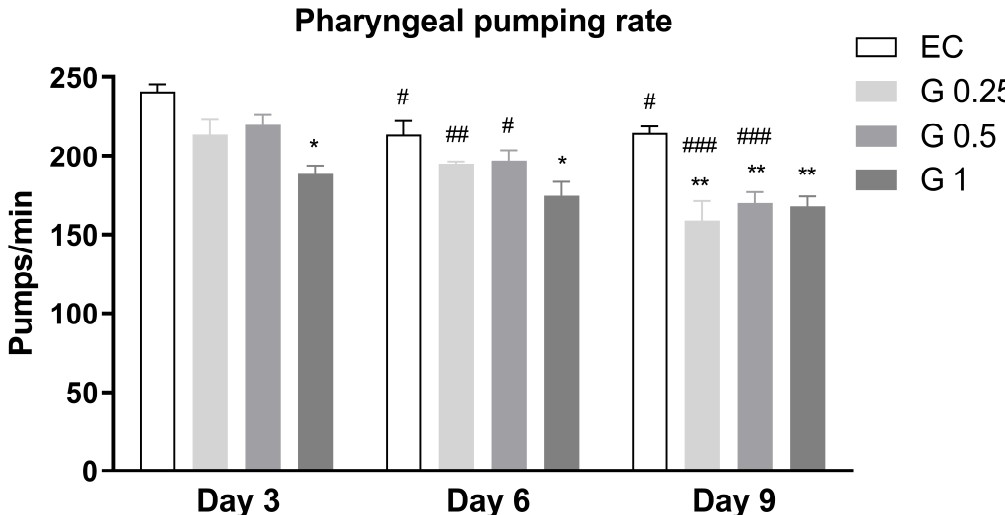

**Figure 2.** Effect of treatment with Terasen® 250 μg/mL (G 0.25), 500 μg/mL (G 0.5), or 1000 μg/mL (G1) and *E. coli* on pharyngeal pumping rate of *C. elegans* in the treated groups. Results are expressed as mean ± SEM. * and ** indicate a significant difference vs. the control group on the same day ($p < 0.05$ and 0.01) while #, ##, and ### indicate a significant difference vs. the same group on day 3 ($p < 0.05$, 0.01, and 0.001). Significance of the factors in Two-Way ANOVA: Treatment—**** ($F_{(3,24)} = 21.48$; $p < 0.0001$); Day—**** ($F_{(2,24)} = 26.88$, $p < 0.0001$); Interaction: ns ($F_{(6,24)} = 1.923$; $p = 0.1181$) (Two-Way ANOVA followed by the post-hoc Tukey's test).

*3.3. Reproduction Assessment*

To investigate the impact of Terasen® on *C. elegans* reproduction, egg-laying was evaluated in animals treated with different concentrations of the formulation (250, 500, or 1000 μg/mL). The results revealed a significant reduction in egg production in animals exposed to Terasen® compared to the control group ($p < 0.001$; Figure 3). It is observed that this drastic decrease tends to decrease from the second day on and could be explained by possible initial stress due to an exogenous compound in the medium or the treatment stress. Still, mainly due to the initial decrease, there was a statistical difference in overall values given by the total progeny count (Area Under the Curve). However, it is crucial to notice that, considering the treated groups only, the progeny had a significant and concentration-dependent increase, suggesting that the decrease compared to the control was not caused by the product per se.

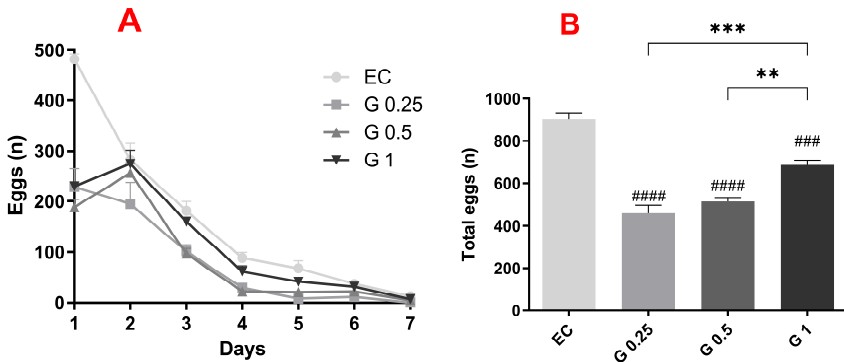

**Figure 3.** Effect of treatment with Terasen® 250 μg/mL (G 0.25), 500 μg/mL (G 0.5), or 1000 μg/mL (G1) and *Escherichia coli* on *C. elegans* reproduction. (**A**) Eggs laid per day (mean ± SEM) of the groups according to the days evaluated. (**B**) Total progeny counted. ** and *** denote differences among treated groups ($p < 0.01$ and 0.001); ### and #### denotes differences vs. the control ($p < 0.001$ and 0.0001) ($F_{(3,16)} = 52.2$; $p < 0.0001$; One-Way ANOVA followed by Tukey's post hoc test).

### 3.4. Locomotion Assessment

The movement capacity of *C. elegans* was evaluated in control animals and animals treated with different concentrations of Terasen® (250, 500, or 1000 µg/mL). The results demonstrate that Terasen® did not promote decline or improvement in nematode motility in the young adult phase (day 4). Although the overall data from Two-Way ANOVA were not statistically significant for the treatment factor considering all days ($p = 0.1509$; Two-Way ANOVA), during the aging phase (day 8), treated animals' motility was reduced compared to the control ($p < 0.05$ in G1; Figure 4).

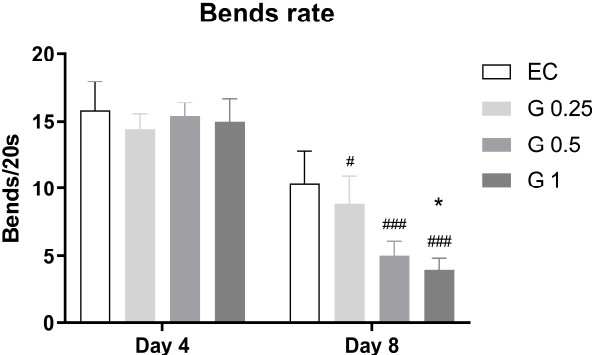

**Figure 4.** Effect of treatment with Terasen® at 250 µg/mL (G 0.25), 500 µg/mL (G 0.5), or 1000 µg/mL (G1) and *E. coli* on the locomotion of *C. elegans*. Results are expressed as mean and SEM, where # and ### indicates a difference vs. the same group on day 4 ($p < 0.05$ and 0.001); * indicates a significant difference compared to the control on the same day ($p < 0.05$). Significance of the factors in Two-Way ANOVA: Treatment—ns ($F_{(3,32)} = 1.891$; $p = 0.1509$); Day—**** ($F_{(1,32)} = 48.78$; $p < 0.0001$); Interaction: ns ($F_{(3,32)} = 1.708$; $p = 0.1850$) (Two-Way ANOVA followed by the post hoc Tukey's test).

### 3.5. Size Evaluation

The results demonstrated a reduction in body length with increasing Terasen® concentration compared to the control group on day 4 (Figure 5). However, on the 8th day of adulthood, with increased exposure time, the length of the animals treated with Terasen® and the control group did not show a noticeable variation when compared to each other. Considering data from all days, however, a significant effect of the treatment was observed ($p < 0.05$; Two-Way ANOVA) that was dependent on the day (Interaction: $p < 0.05$).

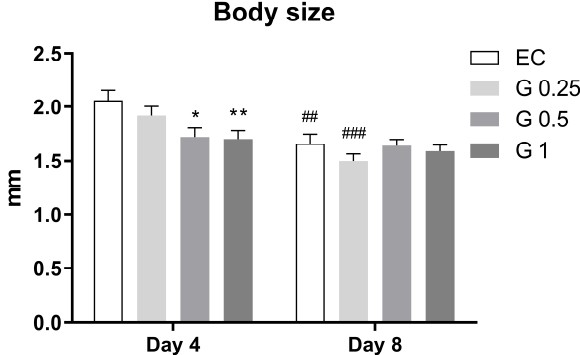

**Figure 5.** Effect of treatment with Terasen® at 250 µg/mL (G 0.25), 500 µg/mL (G 0.5), or 1000 µg/mL (G1) and *E. coli* on the size of *C. elegans*. Results are expressed as mean and SEM, where * and ** indicates a significant difference vs. the control group on the same day ($p < 0.05$ and 0.01); ## and ### indicates a significant difference vs. the same group on day 4 ($p < 0.01$ and 0.001). Significance of the factors in Two-Way ANOVA: Treatment—* ($F_{(3,72)} = 3.091$; $p = 0.0323$); Day—**** ($F_{(1,72)} = 22.00$; $p < 0.0001$); Interaction: * ($F_{(3,72)} = 2.872$; $p = 0.0421$) (Two-Way ANOVA followed by the post hoc Tukey's test).

### 3.6. Lifespan Evaluation

The potential of Terasen® on longevity was assessed. The results showed that, compared to the control group, the average lifespan of *C. elegans* was significantly prolonged after exposure to Terasen® treatment (Figure 6; Table 1). The median survival (days) was 6 for the control, 6 for Terasen® 250 μg/mL, 7 for 500 μg/mL, and 8 for 1000 μg/mL. The maximum lifespan of *C. elegans* increased by 5 days after treatment with 500 μg/mL, and 5 days with 1000 μg/mL of Terasen® compared to the control group.

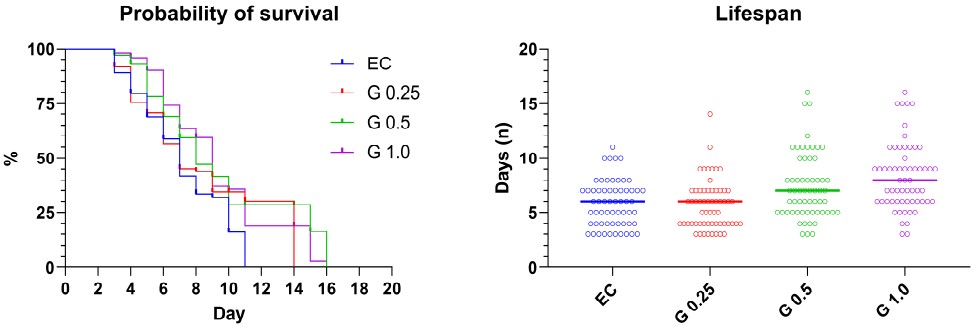

**Figure 6.** Effect of treatment with Terasen® at 250 μg/mL (G 0.25), 500 μg/mL (G 0.5), or 1000 μg/mL (G1) and *E. coli* on the lifespan of *C. elegans*. On the left is shown the probability of survival each day; on the right is shown the individual lifespan values of the worms in each treatment and the median. The log-rank test demonstrated a significant difference among the survival curves (see Table 1 for details).

**Table 1.** Descriptive statistics and results from the survival analysis.

| Group | n | Median [Interquartiles] | Mean ± SEM | Range | Sum | Chi-Square (vs. Control/Total) | Log-Rank $p$ (vs. Control/Total) |
|---|---|---|---|---|---|---|---|
| Control (EC) | 60 | 6 [4/7] | 5.83 ± 0.27 | 8 [3–11] | 350 | - | - |
| Ter 0.25 | 60 | 6 [4/7] | 5.70 ±0.29 | 11 [3–14] | 342 | 1.160 | 0.2814 |
| Ter 0.5 | 60 | 7 [5/9.5] | 7.40 ± 0.40 | 13 [3–16] | 444 | 5.503 | 0.0190 |
| Ter 1.0 | 60 | 8 [6/9.8] | 8.35 ± 0.41 | 113 [3–16] | 501 | 9.943 | 0.0016 |
| Total | 240 | 6 [5/8] | 6.82 ± 0.19 | 13 [3–16] | 1637 | 8.471 | 0.0036 |

### 4. Discussion

With aging, there is an imbalance of reactive species. Hence, we assessed the radical scavenging of Terasen® through the DPPH and ABTS methods. The radical scavenging in the DPPH assay was low compared to the positive control gallic acid; on the other hand, it was high in the ABTS assay. This could be explained by the presence of the phenolic compounds from the formulation with a higher affinity for radical scavenging of ABST [17,18]. Although the antioxidant activity was not shown in vivo, the radical scavenging activity could have a role in the life-extending effect observed.

Then, the potential of Terasen® on longevity was evaluated in vivo using *C. elegans*. *C. elegans* has proven to be a valuable organism for studying aging, as it undergoes noticeable changes over time, such as alterations in body movement, pharyngeal pumping, egg-laying posture, and body size, among others [19,20]. This study observed that the pharyngeal pumping rate of Terasen®-treated groups was reduced compared to the control group, indicating that Terasen® likely decreased the nematode's feeding behavior. In addition, the production of eggs of the nematodes treated with the product decreased compared to the control group. However, when comparing only the groups treated, there was a significant and concentration-dependent increase in the number of eggs.

There were no statistically significant changes in the animals' body movements when considering all days ($p$ = 0.1509; Two-Way ANOVA). However, a relevant difference was observed considering the aging phase only. Next, we assessed the animals' body sizes at the young adult phase (day 4) and the aging phase (day 8). The results showed that

after 4 days of Terasen® treatment, there were significant changes in the body length of the nematodes compared to the control. This difference was mitigated on the eighth day, suggesting that the treatment could delay the animals' growth without affecting the final length. This assumption is supported by the significant ($p < 0.05$) interaction between Day vs. Treatment in Two-Way ANOVA, which means that the effect depends on the day.

Finally, we observed that the treatment significantly increased the average and maximum lifespan of the nematodes in a concentration-dependent manner.

In summary, the in vivo results show mainly that the treatment (1) decreased pharyngeal contractions and hence the feeding behavior of the animals; (2) delayed the body growth without affecting the final size; (3) decreased the body movements in the aging phase, and (4) apparently decreased the number of laid eggs compared to the untreated group. However, this latter effect could be caused by another factor rather than the product itself since, among the treated animals, the number of eggs increased significantly in a concentration-dependent fashion, which is the opposite of what would be expected if the nutraceutical caused the effect.

There are several resemblances of the results with the effects of caloric restriction (CR)—which can also increase the lifespan. However, as discussed, some observed results suggest that caloric restriction alone cannot explain all the observed effects.

First, the feeding behavior (pumping rate) decreased consistently on all days assessed. This is similar to what is observed in *eat-2* mutant worms used as CR models that have increased lifespan [21]. The decreased pumping rate in these mutants is caused by a defective mutation [22], meaning that the feeding decrease is not voluntary. However, it has been shown that the animals' satiety can also affect the pumping rate, as it decreases after feeding [23]. Our results indicate that the latter is the case because it is also supported by the observed trend of decreased movements (body bends) of the treated animals. As previously demonstrated by Lüersen [21], animals under CR have significantly increased locomotor activity compared to ad-libitum-fed worms, which can be an adaptive response in search of food. This suggests that the treatment increased the animals' satiety.

Another similarity with CR was the decreased body size of the animals. However, as mentioned previously, the reduced size was observed only in young adult animals, while the final size was unchanged. This is different from what would be expected under caloric restriction. Mörck [24] showed that CR models using *C. elegans*, in general, have higher discrepancies in sizes in later phases of development compared to earlier stages, in contrast to our results. Finally, since CR decreases the number of eggs laid [25], if we consider that, among the treated groups, the number of laid eggs increased proportionally to the product concentration, this would also be different from what was expected if the increased lifespan were caused through CR.

Studies in the literature report that some plant secondary metabolites can aid in longevity by acting on molecular pathways that modulate stress resistance and metabolic efficiency. In line with the results observed here, some phytochemical compounds present in Terasen® have been reported to increase the lifespan and stress resilience in *C. elegans* and other organism, including anthocyanins [26–32], quercetin [33,34], polyphenols [28,35], carotenoids [36,37], and tocotrienols [38,39].

Despite the particularities of each product tested, these compounds increase the stress resistance by improving the organism antioxidant activity. This is evidenced by decreased levels of ROS and MDA (as seen in [26]). The improved antioxidant defense involves the increased expression of antioxidant enzymes (such as SOD-3 and GST-4, as reported in [34]) mediated by the transcription factors DAF-16 and SKN-1 (orthologues of the human FoxO and Nrf2, respectively). Another pathway involved, mainly in the improved metabolic efficiency, is the insulin/IGF-1 (as seen in [35]). The improved metabolic efficiency, in turn, is responsible for increasing the lifespan through mTOR signaling [40,41].

Although the mechanisms were not assessed here, the increased lifespan observed in vivo is consistent with improved metabolic efficiency, which could also explain the decreased need for feeding observed without impairing important physiological functions.

Further research is warranted to elucidate the mechanisms observed and to demonstrate the antioxidant effect in vivo.

## 5. Conclusions

The results show that the product significantly extended the median and maximum lifespan of the worms compared to untreated animals. Terasen® also affected the feeding behavior of the nematode, decreasing the pharyngeal pumping rate and body movements. Despite some similarities with caloric restriction, some results are not by it (including the animals' final size, decreased body movements, etc.). These results are consistent with other research showing that secondary metabolites found in Terasen® can also prolong life in *C. elegans*.

**Author Contributions:** Conceptualization, J.C.T.C. and E.L.d.M.; methodology, E.L.d.M.; validation, E.L.d.M.; formal analysis, H.d.O.C.; investigation, I.d.S.S., A.L.d.N., B.A.M.T. and N.N.D.C.; resources, J.C.T.C. and C.E.W.; writing—original draft preparation, E.L.d.M.; writing—review and editing, E.L.d.M. and A.C.M.P.; visualization, L.F.M. and A.N.C.; supervision, J.C.T.C.; project administration, E.L.d.M. and J.C.T.C.; funding acquisition, J.C.T.C. All authors have read and agreed to the published version of the manuscript.

**Funding:** This research received National Council for Scientific and Technological Development (CNPq) external funding—CNPq Call—No. 68/2022—Master's and Doctoral Program for Innovation MAI/DAI.

**Institutional Review Board Statement:** Not applicable.

**Informed Consent Statement:** Not applicable.

**Data Availability Statement:** Data will be made available upon reasonable request.

**Conflicts of Interest:** The authors declare no conflict of interest.

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
