# Peer review of "Association of Euterpe oleracea, Bixa orellana, Myciaria dubia, and Astrocaryum aculeatum (the Terasen® Nutraceutical) Increases the Lifespan of Caenorhabditis elegans"

_2673-9259, doi:10.3390/jal3040022_

Round 1

Reviewer 1 Report

Comments and Suggestions for Authors

Please refer to the comments in the attached review document.

Comments on the Quality of English Language

There are some minor mistakes in grammar.

Author Response

The letter is attached. 

Reviewer 2 Report

Comments and Suggestions for Authors

In this manuscript, de Melo et al. analyzed the effects of Terasen on the aging of C. elegans. They found that Terasen had antioxidant activity and influenced the feeding behavior and lifespan of C. elegans. I have some comments:

- in line137-line144, the data of radical scavenging assessment should be displayed in a figure.

- Although it is shown that Terasen exhibited antioxidant activity which may be associated with aging. It is necessary to assess the ROS level in C. elegans.

- in Figure 2B. Why use Area under curve? It would be better to use total number of progenies.

- in Figure 3. the difference between G0.5, G1 and EC seems significant.

- in Discussion line 226-240. These 4 paragraphs seem more suitable to be put in the introduction.

- Since it is speculated that Terasen may activate the caloric restriction or antioxidant pathway, it is necessary to assess whether the longevity effect of Terasen is dependent on DAF-16 or SKN-1, for example using the daf-16 and skn-1 loss-of-function mutants to perform lifespan.

- line 284-300 these paragraphs are repeated.

- Figure 6 seems irrelevant to this study. The role of Terasen should be indicated in this figure.

- in line 317 and 329, "[29] reported..." and "[40] reported..." these sentences should be rephrased. The author name rather than the reference number should be indicated in these sentences.

- in line 320-321, age-1, daf-2 and daf-16 should be italic.

- in line 325, line 331, line 336, line 360, line 366, line 374, and line 378 "...reported by [34].", "...reported by [41].", and "In a study conducted by [43]..." et al... these sentences should be rephrased. The author name rather than the reference number should be indicated in these sentences.

- the Discussion part seems too redundant. it should be refined.

Comments on the Quality of English Language

The English language should be polished. 

Author Response

The letter is attached. 

Reviewer 3 Report

Comments and Suggestions for Authors

Line 28: I disagree with a “remarkable ability to extend” lifespan. I agree that your data seems to support a positive effect on lifespan, but I think remarkable is an exaggeration.  

Line 157: Escherichia coli should be E. coli here, and other places in the manuscript, and should be italicized.  

Line 183: what do you mean by “considering the data from all days?” As you mentioned for day. It seems as though this mixture has a negative effect on locomotion.

Line 317: I don’t think you should start a sentence with a reference like this. Write it as “Li et al., [29] …”

It looks like lines 267-283 is duplicated in lines 284-300.

Discussion should be improved. These is a lot of content regarding the use of worms. Should focus more on Terasen and how it might work.

Extract appears to have a negative effect on eating (pharyngeal pumping), locomotion, and egg laying. A simple explanation is that the mixture tastes bad, and the worms are eating less, resulting in decreased egg production and life extension, and possibly decreased locomotion. I think you need show whether this mixture can extend lifespan in an eat-2 background.

If this mixture is working by scavenging radicals, then it should confer protection against as free radical generator such as paraquat. This experiment should be done.

You seem to suggest that caloric restriction may be involved in the action of Terasen. The effect of Terasen should be tested in a daf-2 background.

Author Response

The letter is attached. 

Reviewer 4 Report

Comments and Suggestions for Authors

In the present study the Authors aimed to evaluate the impact of Terasen®, a nutraceutical containing standardized extracts of Euterpe oleracea, Myrciaria dubia, and purified oil of Bixa orellana and Astrocaryum aculeatum on the lifespan of Caenorhabditis elegans, a widely used model organism for aging research. The manuscript is written clearly. The data are presented clearly and justify the conclusions.

Minor remark:

-        Page 9, lines 317-319- the following statement needs to be clearly written: “[29] reported that the extract of black rice, which is a rich source of anthocyanins (representing 43% of the extract), extended the lifespan, enhanced stress resistance, increased antioxidant enzymes activity, and reduced the accumulation of lipofuscin, ROS, 319 and MDA”.

Author Response

The letter is attached. 

Round 2

Reviewer 1 Report

Comments and Suggestions for Authors

I believe the authors have addressed most of the issues and questions raised by me and the other reviewers. I suggest that the authors test the potential pseudo-CR effect with relevant mutants before proceeding to the next steps.

Author Response

We thank the referee’s valuable suggestion for our study. As we write, we are soughing to get the resources to work with mutant worms, and all comments given here will be considered.

Reviewer 2 Report

Comments and Suggestions for Authors

The total sample size of lifespan (n=20) is too small. For each independent lifespan experiment, the sample size should be at least 50 worms per condition. And it is typical to set up about 100 worms per condition. How many independent lifespan experiments have you performed? Data of 3 independent lifespan experiments should be included to support you conclusion.

Comments on the Quality of English Language

English language seems ok.

Author Response

In the first version, we performed three independent experiments (three dishes) with 20 animals in each group, totaling 60 animals per treatment. Then, we averaged the number of worms by three and considered alive the lowest integer of this division. We reanalyzed the data, considering each worm as a separate n (n = 60 per group). 

Reviewer 3 Report

Comments and Suggestions for Authors

I'm good. I feel that you have either addressed my concerns or attempted to do so to the best of your ability. 

Author Response

We thank the referee for the valuable feedback.